# Frequency-Specific Analysis of the Dynamic Reconfiguration of the Brain in Patients with Schizophrenia

**DOI:** 10.3390/brainsci12060727

**Published:** 2022-06-01

**Authors:** Yanli Yang, Yang Zhang, Jie Xiang, Bin Wang, Dandan Li, Xueting Cheng, Tao Liu, Xiaohong Cui

**Affiliations:** College of Information and Computer, Taiyuan University of Technology, No. 209, Daxue Street, Yuci District, Jinzhong 030024, China; yangyanli01@tyut.edu.cn (Y.Y.); zy_tyut_edu@163.com (Y.Z.); xiangjie@tyut.edu.cn (J.X.); wangbin01@tyut.edu.cn (B.W.); lidandan@tyut.deu.cn (D.L.); c15630859037@163.com (X.C.); 17865318631@163.com (T.L.)

**Keywords:** frequency-specific, multilayer network, dynamic reconfiguration

## Abstract

The analysis of resting-state fMRI signals usually focuses on the low-frequency range/band (0.01–0.1 Hz), which does not cover all aspects of brain activity. Studies have shown that distinct frequency bands can capture unique fluctuations in brain activity, with high-frequency signals (>0.1 Hz) providing valuable information for the diagnosis of schizophrenia. We hypothesized that it is meaningful to study the dynamic reconfiguration of schizophrenia through different frequencies. Therefore, this study used resting-state functional magnetic resonance (RS-fMRI) data from 42 schizophrenia and 40 normal controls to investigate dynamic network reconfiguration in multiple frequency bands (0.01–0.25 Hz, 0.01–0.027 Hz, 0.027–0.073 Hz, 0.073–0.198 Hz, 0.198–0.25 Hz). Based on the time-varying dynamic network constructed for each frequency band, we compared the dynamic reconfiguration of schizophrenia and normal controls by calculating the recruitment and integration. The experimental results showed that the differences between schizophrenia and normal controls are observed in the full frequency, which is more significant in slow3. In addition, as visual network, attention network, and default mode network differ a lot from each other, they can show a high degree of connectivity, which indicates that the functional network of schizophrenia is affected by the abnormal brain state in these areas. These shreds of evidence provide a new perspective and promote the current understanding of the characteristics of dynamic brain networks in schizophrenia.

## 1. Introduction

As a complex psychotic disorder, schizophrenia (SZ) usually occurs in adolescence or adulthood and is related to abnormal integration between distal brain regions [1]. It is considered a chronic and diverse genetic disease with abnormal perception, emotion, and brain connections [2]. Clinically, it is emotional and cognitive dysfunction, accompanied by symptoms such as hallucinations and delusions [1,3,4]. Therefore, SZ is increasingly considered a disease caused by brain network dysfunction.

In functional magnetic resonance imaging (fMRI) studies, traditional brain network construction methods are generally during the resting-state. Current dynamic network analyses have confirmed that fluctuations in functional connections exist, which has attracted increasing attention in the academic world [5,6]. Studies have shown that dynamic network analysis can better detect fluctuations in the brain’s functional connections. For example, Gifford and colleagues used a modeling method called the dynamic modular organization to investigate better inter-group differences in dynamic community structure in SZ [7]. Cui and colleagues also believe that dynamic functional connectivity can reflect the time-varying characteristics of brain networks and capture the changes of network topology and cognitive behavior over time [8]. The most common method of dynamic network research is to use sliding window technology to divide blood oxygen level-dependent (BOLD) signals into shorter time intervals or windows and obtain functional brain networks from each time interval [9,10]. These intervals provide more reliable results for the dynamic analysis of SZ.

Dynamic reconfiguration of multilayer brain networks can be effective to analyze SZ caused by cognitive impairment and is increasingly used to find abnormal neural activities. Previous studies have revealed that low frequency is generally associated with abnormal dynamic functional connections of the brain network. To illustrate, Debo Dong and colleagues [11] comprehensively studied the dynamic reconfiguration of the dynamic reorganization of resting-state functional connectivity (rsFC) in SZ when the brain frequency was less than 0.1 Hz and found patients who showed consistently increased rsFC variability in sensory and perceptual systems. Xiao Wang and colleagues believed there are significant differences between slow4 and slow5’s SZ functional connectivity densities [12]. In conclusion, studies on the dynamic reconfiguration of multilayer brain networks will provide evidence for further exploration of the mechanism of schizophrenia.

Previous studies on SZ’s multilayer dynamic brain network generally focused on low frequency [12,13] because this frequency band is usually associated with the abnormal dynamic functional connection of the brain network. Still, it cannot completely cover the complex neural activities in the resting-state brain. Some research studies have shown that brain regions in different frequency bands have different resting-state functional connections. In other words, physiological signals in the same brain network may compete or cooperate with each other in various frequency bands [1]. Recently, researchers have found that the high-frequency band can also provide valuable information for the diagnosis of SZ. For example, Gifford and colleagues found that the flexibility of a satisfactory score is higher at 0.08–0.25 Hz, which supports the view that SZ involves dynamic brain network changes [7]. Yu et al. and colleagues showed that considering different high-frequency bands helps measure brain activity in SZ [14]. That is to say, frequency is closely related to the study of SZ and the abnormal structure of large-scale brain networks.

Based on the above research, we assumed that the brain network connection of SZ is frequency dependent. Therefore, we used the following five different frequency bands to explore the changes between brain regions (full frequency (0.01–0.25 Hz), slow5 (0.01–0.027 Hz)), slow4 (0.027–0.073 Hz), slow3 (0.073–0.198 Hz) and slow2 (0.198–0.25 Hz). A multilayer community detection algorithm is applied to identify the temporal community and calculate recruitment, integration, and module allegiance [15]. These two indicators reflect the dynamic interactions between and within different brain regions and quantify community changes over time [8] (Figure 1). In conclusion, our purpose was to study the dynamic reconfigurability of the brain network structure of SZ in different frequency bands and provide new ideas for future research.

## 2. Materials and Methods

### 2.1. Participants

The data were selected from the UCLA Consortium for Neuropsychiatric Phenomics LA5c Study project, and the UCLA Institutional Review Board approved the study. The data were obtained via a public database, open fMRI (https://openfmri.org/dataset/ds000030/, accesssed on 1 September 2020). Our study comprised 82 subjects, including 42 schizophrenia patients (SZ) and 40 normal controls (NC). There was no significant difference in age and gender between the NC and SZ (T-test). It is worth noting that 42 SZ in 82 subjects participated in the evaluation of SAPS. Specific demographic characteristics are shown in Table 1.

### 2.2. Imaging Acquisition and Preprocessing

All subjects underwent structural and functional MRI scans on a 3-T scanner with a 32 channel head coil at the China University of Electronic Science and Technology. Soft foam and earplugs were used to fix the subjects’ head and reduce scanning noise during the scanning process. They were asked to stay relaxed, open their eyes, and avoid intentional thinking activities. Resting-state fMRI data were collected using a sequence of T2*-weighted echo-planar imaging (EPI). The parameters are as follows: repetition time (TR) = 2 s, echo time (TE) = 30 ms, slice thickness = 4 mm, slices = 34, flip angle = 90°, field of view (FOV) = 192 mm, and matrix = 64 × 64. The resting-state fMRI scan lasted a total of 304 s. For the structural scan, a T1-weighted high-resolution anatomical scan was collected with the following parameters: slice thickness = 1 mm, slices = 176, repetition time (TR) = 1.9 s, echo time(TE) = 2.26 ms, matrix = 256 × 256, and field of view (FOV) = 250 mm.

Data preprocessing was carried out by the DPABI toolbox (http://rfmri.org/dpabi, accessed on 1 September), the first ten volumes of the signal were discarded, and the data of the other 142 time points were preprocessed as follows: (1) slice timing correction; (2) realignment; (3) normalization: the image space was standardized to the Montreal Neurological Institute (MNI) head anatomy template and resampled with 3 × 3 × 3 mm^3^ voxels [16]. (4) Filtering: considering the frequency-specific, five frequency bands are divided here: full frequency (0.01–0.25 Hz), slow5 (0.01–0.027 Hz), slow4 (0.027–0.073 Hz), slow3 (0.073–0.198 Hz), and slow2 (0.198–0.25 Hz) [17]; (5) Smoothing: the images were spatially smoothed using a Gaussian filter with a full-width at half-maximum (FWHM) of 6 mm [18]; (6) the covariates were removed, and the brain was divided into 90 regions using the automatic anatomical marker template (AAL) [19] and residual time-series were extracted for each voxel.

According to the time series extracted by the AAL template, we divided brain regions into five functional networks [20]: somatosensory/motor and auditory network (SMN), visual network (VN), attention network (AN), default mode network (DMN) and limbic/paralimbic and subcortical network (LSN).

### 2.3. Formatting of Mathematical Components

After the above processing, each subject was divided into five frequency bands. The sliding time window is divided for each frequency band, and the time series is divided into smaller time intervals. According to Leonardi and Van De Ville [21], the minimum window size that can be used in dynamic network studies is $1/f_{min}$ ($f_{min}$ denotes the minimum frequency of the data included). Here, we select the minimum value of the whole frequency band (0.01 Hz) as the frequency for dividing the time window. Hence, the window length we used was 100 s (50 TRs). Each window was shifted 2 s (1 TR), resulting in 93 overlapping windows.

To better consider the changes in network organization over time, we introduced a multilayer network model [22]. We connected nodes in one time window to themselves in adjacent time windows to represent time dependence. The supra-adjacency matrix of an f-layered multilayer network can be expressed as:(1)$$A=\left[\begin{array}{ccc} A_{1} & \cdots & H_{1f}\\ \vdots & \ddots & \vdots \\ H_{f1} & \cdots & A_{f} \end{array}\right]$$
where $A_{\alpha }$ is the intra-layer network adjacency matrix of layer α, 1≤α≤f. $H_{kl}$ is a diagonal matrix and a square matrix corresponding to the inter-layer connection matrix between layers k and l.

### 2.4. Multilayer Community Detection

The community detection algorithm provides a method called “module”, which can decompose the network into dense node groups. Here, we use a multilayer community detection algorithm called Genlouvain to identify the brain community of each participant [23,24]. It divides the communities in the multilayer network by optimizing the multilayer modular quality function Q and getting the information about the nodes. It can be defined as:(2)$$Q=\frac{1}{2\mu }\sum ijlr\{(A_{ijl}-\gamma_{l}B_{ijl})\delta_{lr}+\delta_{ij}\omega_{jlr}\}\delta (g_{il},g_{jr})$$
where μ is the total edge weight of the network and $\delta (g_{il},g_{jr})$ is 1 if nodes belong to the same community and 0 otherwise [25]. $A_{ijl}$ represents the adjacency matrix between node i and node j in layer l, $\gamma_{l}$ is the structural resolution parameter of layer l and element $B_{ijl}$ represents the component of layer l matrix corresponding to the optimized zero model. The element $\omega_{jlr}$ gives the connection strength from node j of layer r to node j of layer l, which is called “inter-layer coupling parameter”. Here, we set the structure resolution parameter and the inter-layer coupling parameter to the default value of 1 [8,26,27].

We used the Genlouvain MATLAB toolbox [23,24] to calculate the community allocations. This community assignment represents the evolution of assumed functional modules in the brain network over time. In addition, the algorithm is a generalized greedy-like algorithm [28], whose output may change at each run due to the random nature of the optimized partition function. Therefore, we repeat the detection algorithm 50 times, and then take the average of these measures as the final estimate value.

### 2.5. Dynamic Network Statistics

#### 2.5.1. Module Allegiance

In the module allegiance matrix P, each element $P_{ij}$ represents the probability that node i and node j are assigned to the same community during the whole scanning process [29]. The matrix P consists of *N* × *N* and is the number of brain regions (In this study, *N* = 90). $P_{ij}$ can be written as follows:(3)$$P_{ij}=\frac{1}{OT}\overset{O}{\underset{o=1}{\sum }}\overset{T}{\underset{t=1}{\sum }}a_{i,j}^{k,o}$$
where O is the number of iterations of the multilayer community detection algorithm, T is the number of layers. For each optimization O and layers T, if they are in the same community network, the value of module loyalty is 1 (the values on the main diagonal of the matrix are all 1); otherwise, it is 0.

#### 2.5.2. Recruitment and Integration

Recruitment: We use the module allegiance matrix to evaluate the dynamic role of cognitive systems in task execution [15]. According to the above module allegiance matrix, recruitment and integration are calculated. Recruitment is defined as the probability that its region and the region from the same system appear in the community together. For node i in the community network S, the recruitment is specifically expressed as follows:(4)$$R_{i}^{S}=\frac{1}{n_{s}}\underset{j\in S}{\sum }P_{ij}$$
where $n_{S}$ is the number of nodes in network S. $P_{ij}$ represents the number of times that nodes i and j are assigned to the same module.

Integration: Integration is defined as the probability that its region will co-appear in the community with regions from other systems. For node i in the community network S, the integration can be written as follows:(5)$$I_{i}^{S}=\frac{1}{N-n_{S}}\underset{j\notin S}{\sum }P_{ij}$$
where N is the total number of brain regions.

### 2.6. Statistical Analysis

An independent sample T-test was used for statistical tests to quantify the differences in integration and recruitment between NC and SZ. For each metric, we averaged the metrics obtained from the network. Spearman’s correlation was used to assess the correlation between SZ and SAPS scale scores. The Benjamini and Hochberg error discovery rate (BH_FDR) method was used to calibrate all the results, and the threshold value for the significant difference was set to <0.05.

## 3. Results

### 3.1. Group Comparisons of the Whole-Brain Level at Different Frequencies

In this part, we average the recruitment and integration of the whole brain region of each subject and discuss the dynamic difference between brain networks in full frequency and slow5, slow4, slow3 and slow2. See Table 2 for the specific number of modules As shown in Figure 2A, there was no significant difference in the integration between the two groups, but the recruitment decreased significantly in full frequency (T(80) = −2.600, P = 0.011) and slow3 (T(80) = −3.090, P = 0.003) (Figure 2B).

### 3.2. Group Comparisons of RSN Level at Different Frequencies

According to the AAL template, we divide 90 brains into five functional networks in the preprocessing. Here, we will study the differences in functional networks of SZ in five frequency bands. As shown in Figure 3A, by comparing the integration between NC and SZ, we found no significant difference in the 0.01–0.25 Hz and slow2, slow4, slow5. However, in slow3, the VN results showed a significant increase (T(80) = 2.263, P = 0.026). As shown in Figure 3B, the recruitment results showed a significant difference in the 0.01–0.25 Hz and slow3.In 0.01–0.25 Hz, the recruitment of VN (T(80) = −2.840, P = 0.006) and AN (T(80) = −2.392, P = 0.019) decreased. In slow3, except for SMN, the recruitment of the other four RSNs are significant. Among them, the recruitment of VN (T(80) = −4.101, P = 0.000) and AN (T(80) = −3.577, P = 0.001) decreased significantly. The recruitment of DMN (T(80) = −2.178, P= 0.032) decreased, while that of LSN (T(80) = 2.456, P = 0.016) increased. The specific brain network information is shown in Table 3.

### 3.3. Group Comparisons of RSN to RSN Integration at Different Frequencies

After proving the abnormality of SZ at the RSN level, we are still interested in the details between networks. To further investigate the group differences between SZ and NC in five functional networks, we decomposed these values into the RSN × RSN matrix of mean module allegiance. The results showed that the SZ recruitment is lower than that of NC in the full frequency. In slow3, only the recruitment of SZ with LSM is higher than that of NC, and the other modules are lower than that of NC. In addition, we also observed that both SZ and NC prefer to arrange nodes in their functional network rather than through the whole system; that is, they have the characteristics of high recruitment and low integration (Figure 4).

After knowing that the integration of VN is obvious, we still study which pair of networks has a significant difference in integration. In slow3, it is found that the integration between VN and AN (t (80) = 3.429, P = 0.001) (Figure 5A), VN and DMN (t (80) = 3.000, P = 0.004) (Figure 5B) is more obvious in the five brain networks of RSN to RSN after FDR correction. Moreover, the scores of SZ are higher than NC. The specific RSN to RSN integration information is shown in Table 4. (All information values can be viewed in Table 5).

### 3.4. Group Comparisons of Node Level at Different Frequencies

In addition to the results on the whole brain and RSN levels, we also compared the differences in integration and recruitment at the node level between the NC and SZ. When controlling for multiple comparisons using FDR correction, for integration, in 0.01–0.25 Hz, no significant node differences were observed (Figure 6A); in slow3, they were significantly different in four RSNs modules with a total of sixteen brain regions (Figure 6A). Specifically, the SMN has eight regions. The VN has three regions. The DMN has two regions, and LSN has three regions (Table 6). For recruitment, in 0.01–0.25 Hz, there were found two RSNs modules with a total of six brain regions (Figure 6B). Specifically, the VN has four regions, and LSN has two regions; in slow3, there were thirty-three brain regions in five RSNs modules that were significantly different (Figure 6B). Specifically, the SMN module has one region. The VN has four regions, AN has thirteen regions, DMN has five regions, and LSN has ten regions. The specific node information is shown in Table 7.

### 3.5. Correlation between Network Measures and SAPS Scores

In order to further understand whether integration and recruitment are related to the severity of SZ, we investigated the association between Scale for Assessment of Positive Symptoms (SAPS) and indicators. In the slow3, based on the Spearman correlations, we found there was a negative correlation between the average recruitment score and the SAPS score in the whole brain level (r = −0.305; *p* = 0.05) (Figure 7A). In RSN level, there was a negative correlation between the AN and SAPS score (r = −0.305; *p* = 0.05) (Figure 7B). In node level, there were significant negative correlation between four nodes: inferior frontal gyrus, triangular part (IFGtriang. L) (r = −0.378; *p* = 0.014), inferior frontal gyrus, triangular part (IFGtriang. R) (r = −0.338; *p* = 0.029)) and SAPS score (Figure 7C), middle front gyrus (MFG. L) (r = −0.365; *p* = 0.017), middle frontal gyrus (MFG. R) (r = −0.319; *p* = 0.04) (Figure 7D).

## 4. Discussion

To study the frequency-specific dynamic reconfiguration in SZ, we use a sliding time window to construct multilayer brain networks for each band, respectively. Recruitment and integration are introduced to study the differences between NC and SZ. The results showed that compared with the NC, the dynamic reconfiguration of the whole brain in the full frequency of SZ decreased. A more significant decrease is shown in slow3, which means the dynamic reconfiguration of the multilayer brain network of schizophrenia has frequency specificity. These findings may provide a new perspective for explaining the underlying pathological mechanism of SZ.

### 4.1. Reduced Recruitment in SZ Patients

In terms of whole-brain, the recruitment of SZ decreased significantly in the frequency bands of 0.01–0.25 Hz and slow3. Our results showed that the recruitment ability of SZ is reduced, which affects the cognitive ability of patients. Moreover, this decline can be explained by flexibility. Studies have shown that the dynamic brain network reconfiguration of SZ shows higher flexibility [7,8,9]. In a survey of the genetic risk of dynamic brain network reconfiguration in SZ, this higher flexibility was taken as a manifestation of the lack of organization and stability in the patient’s network module organization [9]. These will lead to instability in SZ.

In addition, at the whole-brain level of the slow3, recruitment is negatively correlated with the SAPS score. This imbalance indicates that the frequency band may regulate the dynamic abnormality of SZ, and the network is disintegrated in these patients, which reduces the alliance preference within the system and the communication between the systems. Besides, SAPS is mainly used to evaluate the positive symptoms of schizophrenia, including hallucinations, delusions, and other symptoms. The higher the SAPA score, the more serious the SZ illness and the worse the SZ cognitive ability.

### 4.2. Abnormal Brain Networks/Regions of Dynamic Reconfiguration in SZ Patients at slow3

Our results also compared several groups of dynamic network reconfiguration at the RSN level. Firstly, for visual network (VN), we found that the recruitment decreased, and the integration increased at the slow3. The evidence of continuous fusion shows that the collection and transmission of high-frequency visual information may contribute to the bottom-up information integration of the brain [1]. According to previous studies, the reduced activation of the region in the visual magnetic plasma path is associated with deficits in motor processing in SZ [30]. Furthermore, visual stimulation has been proved to increase the correlation between functionally related regions and decrease the correlation between unrelated regions simultaneously [31]. In addition, we found that LING and FFG nodes were remarkably correlated in the slow3, indicating that these nodes were more sensitive to the severity of SZ. Lying between the calcarine sulcus, the LING, alone with the FFG [32] next to it, is directly connected to limbic/paralimbic and subcortical network (LSN) [33,34]. The study of SZ found that the LING is a brain region supporting visual memory [35], and the FFG is related to the identification of facial information or other objects [36,37]. JY Jung and colleagues [34] found abnormal filtering of irrelevant information in visual cortices and altered functional connectivity between the frontoparietal network and visual cortices in SZ. Damage to this area can lead to visual memory dysfunction and visual edge disconnection syndrome. Based on these findings, we believe that the abnormal state of the visual cortex will affect the cognitive function of SZ and lead to a mental defect.

Secondly, for attention network (AN), we find that the recruitment falls off at the slow3, but the integration between attention networks and visual networks is more pronounced. Some studies have shown that the reorganization of brain network modules might contribute to attention processing [38], and visual processing and attention deficit are the main causes of disability in SZ [39]. As a part of the attention network, MFG plays a vital role in the dynamic network changes involved in attention processing. Some studies have shown that changing the critical node MFG of SZ affecting the time-varying brain network will reconfiguration brain network modules [40]. Penghui and colleagues found that the MFG interconnected network in the left hemisphere of SZ plays a leading role in visual and attention network topology [38]. In other words, MFG will promote the network connection between the attention network and other regions to improve the ability of continuous attention. We also found notable differences in recruitment between node MFG and IFG, and there was a negative correlation with SAPA scores. The regions of MFG and IFG are usually related to decision making, action inhibition, and conflict monitoring [41]. In SZ, the IFG plays a crucial role in executive functions such as cognitive inhibition and semantic and linguistic functions [42,43] and is associated with language production and linguistic working memory [44]. We believe that the impairment of brain function in these regions will affect the impairment of the functional cortical network. The resulting brain dysfunction may lead to abnormal clinical and cognitive measures [45].

Thirdly, for the default mode network (DMN), we find that the recruitment decreased at the slow3. In addition, our results also indicate the importance of visual network and default mode network in the relationship between dynamic brain networks. DMN, as a distributed network of brain regions [46], and has a high degree of functional connectivity. In SZ, DMN is usually over-activated and over-connected, leading to cognitive impairment, hallucination, and delusions [47]. Susan and colleagues [47] suggest that the uniqueness of neuropsychiatric disorders may reflect the interaction between DMN and other brain networks. Researchers have also found that the DMN regularly disintegrates into many components in the resting-state, which can act synchronously with the sensorimotor and attention networks [5]. Therefore, we believe that the DMN is easily connected with other brain regions in dynamic reconfiguration.

Finally, for limbic/paralimbic and subcortical network (LSN), we find that the recruitment of node DCG and AMYG in slow3 are significantly correlated. DCG is a significant component of the limbic system and is associated with memory and spatial orientation [48]. This indicates that the decrease in connectivity strength between DCG brain regions in psychiatric patients may affect brain memory. AMYG is nestled in the medial temporal lobe, located anterior to the hippocampus [49]. As a part of the limbic system, AMYG is an all-cortical region involved in emotional processes, learning, and memory. The AMYG plays an essential role in social information processing, particularly in associating emotional salience to sensory stimuli [50] through connections with multiple brain regions such as the prefrontal cortex, motor, and sensory regions [51,52]. We believe that the abnormal association between the limbic system and other systems may lead to the pathophysiology and symptomatology of SZ. Ho and colleagues demonstrated that amygdala orbitofrontal functional connectivity decreased in schizophrenic patients in the study of resting functional connectivity [49].

### 4.3. Frequency-Specificity of Multilayer Brain Networks in SZ Patients

Our results showed that the dynamic reconfiguration of multilayer brain networks of SZ is different in full frequency. Moreover, the difference in slow3 is more evident than that in full frequency, and some information which is not observable in the full frequency band can even be observed. Therefore, we speculate that the brain network reconstruction of schizophrenia is related to frequency. Previous studies have indicated that higher low-frequency signals may improve the stability of supervisory regulation during the processing of information and high-frequency signals, and frequency band information should be considered in future studies on SZ [1]. Additionally, previous studies on SZ focused on low-frequency (0.01–0.1 Hz) [12,14], which may ignore the information on high-frequency dynamic weighted functional connectivity. Recent neuroimaging studies have also shown that high-frequency (>0.1 Hz) fMRI signals also alter spontaneous neural activity [53].

We suggest that functional integration between brain regions may occur in a specific frequency domain based on these results. Research Indicators (recruitment and integration) are also more observable in specific frequency domains. Suril R. Gohel’s and colleagues’ results show that the functional integration between brain regions measured by BOLD signal correlation occurs over a wider frequency band than was examined in previous studies. This functional integration between brain regions within the same network is specific to one frequency band [54]. This is consistent with our results and may provide new ideas for future SZ research.

By studying the characteristics of each frequency band, we found that the slow3 was more pronounced for SZ. At the same time, Wang and colleagues obtained higher accuracy in slow3 by using the hierarchical sparse learning method to diagnose schizophrenia (SZ) in resting-state functional magnetic resonance imaging (rs-fMRI) [3]. Besides, Zuo and colleagues analyzed the classification weight distribution and concluded that slow3 had the best classification performance while slow2 had the worst [4]. It is considered that the signals in slow3 and slow2 mainly reflect white matter signals and high-frequency physiological noise, while the signs in slow4 and slow5 mainly reflect gray matter signals [55]. Therefore, we believe that slow3 has a specific research value. This proved our conjecture.

## 5. Conclusions

This study analyzed the dynamic reconfiguration of multilayer brain networks in SZ and normal controls at different frequencies. We found that SZ showed differences in full frequency, which was more significant in slow3. The functional integration between brain regions in SZ was more likely to occur in specific frequency domains. We also found that the recruitment of SZ was significantly abnormal. The decrease in the visual network (VN) and dynamic functional connectivity strength of the attention network (AN) in SZ can decrease the internal links in patients. And this may explain why the RSN level mainly occurred in VN and AN. However, the integration abnormalities in SZ at the RSN level primarily focus on VN. This may be because the collection and transmission of high-frequency visual information contributes to the bottom-up information integration of the brain. In the future, our results may provide potential implications for exploring the neuropathological mechanisms of SZ.

## Figures and Tables

**Figure 1 brainsci-12-00727-f001:**
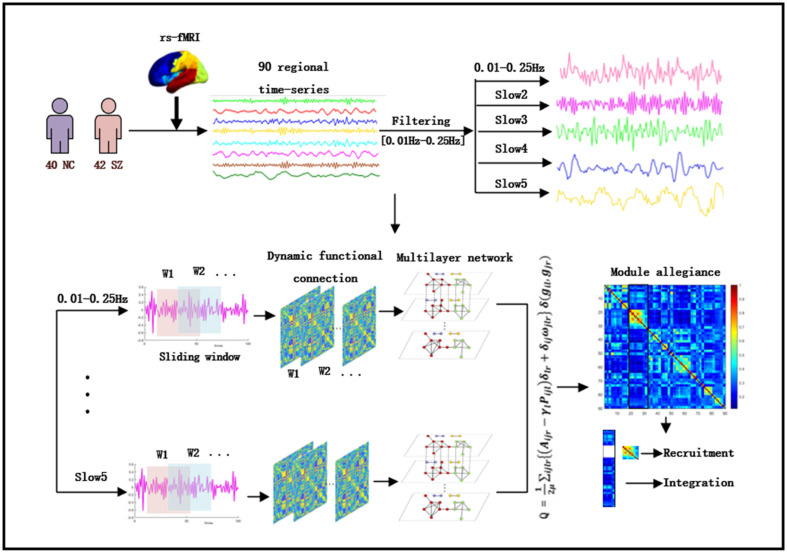
Schematic overview of analysis strategy. The resting-state fMRI data preprocessing results were first divided into 90 brain regions using the existing standard brain profile in the AAL template. Each brain region represented a node in the network. The brain’s physiological signals are decomposed into five frequency bands. The sliding window technique divides the time series into shorter time intervals. The functional connections within each layer are estimated using Pearson correlation. They connect the same nodes in adjacent periods and build a multilayer network for each participant. The dynamic community structure is detected by maximizing the multilayer modular quality function. We calculated the module allegiance matrix, recruitment, and integration to analyze the difference between NC and SZ.

**Figure 2 brainsci-12-00727-f002:**
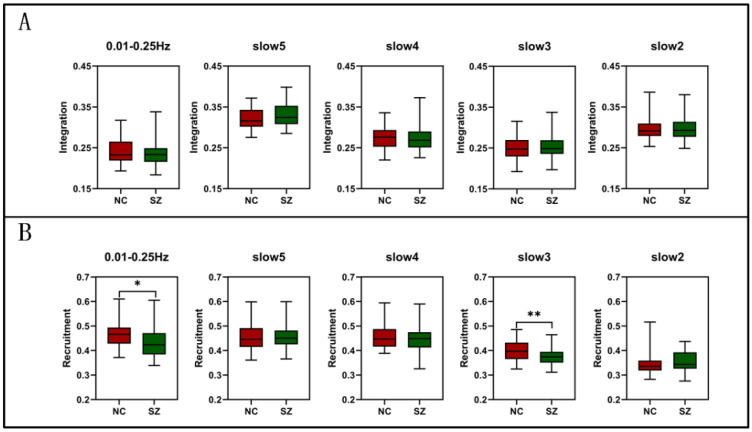
The difference of recruitment and integration from full frequency (0.01–0.25 Hz), slow5 (0.01–0.027 Hz), slow4 (0.027–0.073 Hz), slow3 (0.073–0.198 Hz), and slow2 (0.198–0.25 Hz) in the whole brain level between NC and SZ. (**A**) Integration. (**B**) Recruitment. Asterisk indicates pairwise group differences; * represents *p* < 0.05; ** represents *p* < 0.01.

**Figure 3 brainsci-12-00727-f003:**
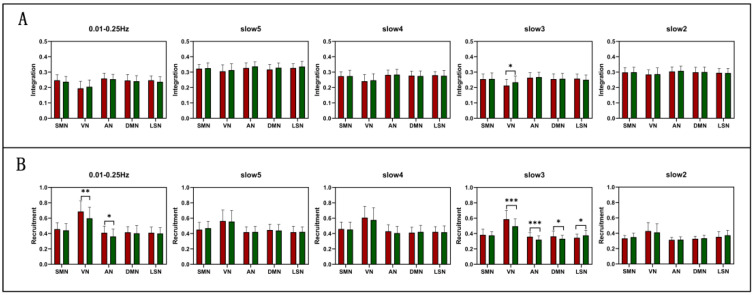
The difference of integration and recruitment from full frequency (0.01–0.25 Hz), slow5 (0.01–0.027 Hz), slow4 (0.027–0.073 Hz), slow3 (0.073–0.198 Hz), and slow2 (0.198–0.25 Hz) in the RSN level between NC and SZ. (**A**) Integration. (**B**) Recruitment. Asterisk indicates pairwise group differences; *** represents *p* < 0.001, ** represents *p* < 0.01, * represents *p* < 0.05.

**Figure 4 brainsci-12-00727-f004:**
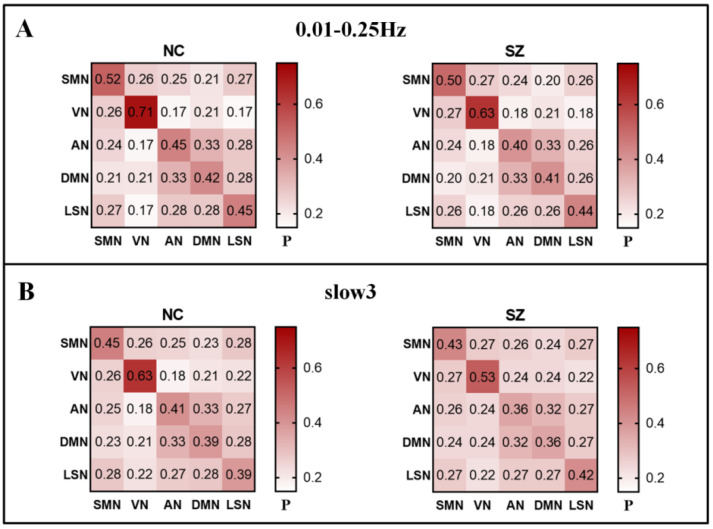
Matrix of module allegiance in NC and SZ. (**A**) full frequency (0.01–0.25 Hz). (**B**) slow3 (0.073–0.198 Hz). P represents the mean value of module allegiance. The main diagonal represents the recruitment coefficient, and the upper/lower triangle represents the integration coefficient.

**Figure 5 brainsci-12-00727-f005:**
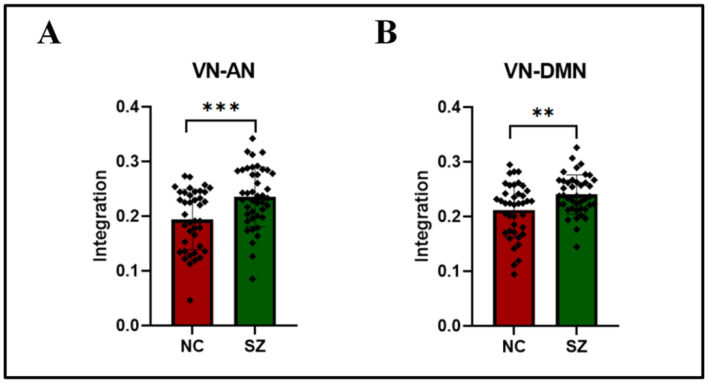
Group differences of integration for each pair of RSN to RSN in slow3 (0.073–0.198 Hz). (**A**) the integration between Visual network(VN) and Attention network(AN). (**B**) the integration between Visual network(VN) and Default mode network(DMN). Asterisk indicates pairwise group differences; *** denotes *p* < 0.001, ** denotes *p* < 0.01.

**Figure 6 brainsci-12-00727-f006:**
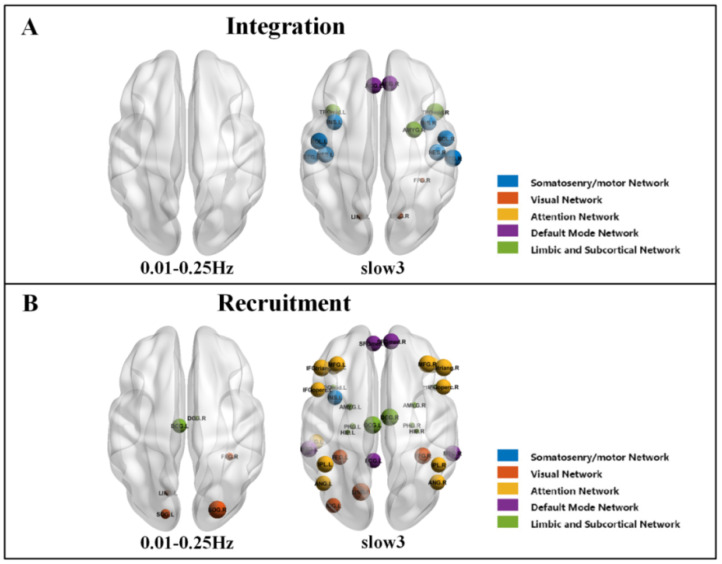
Difference between the NC and SZ in node vulnerability at full frequency (0.01–0.25 Hz) and slow3 (0.073–0.198 Hz). (**A**) Integration. (**B**) Recruitment.

**Figure 7 brainsci-12-00727-f007:**
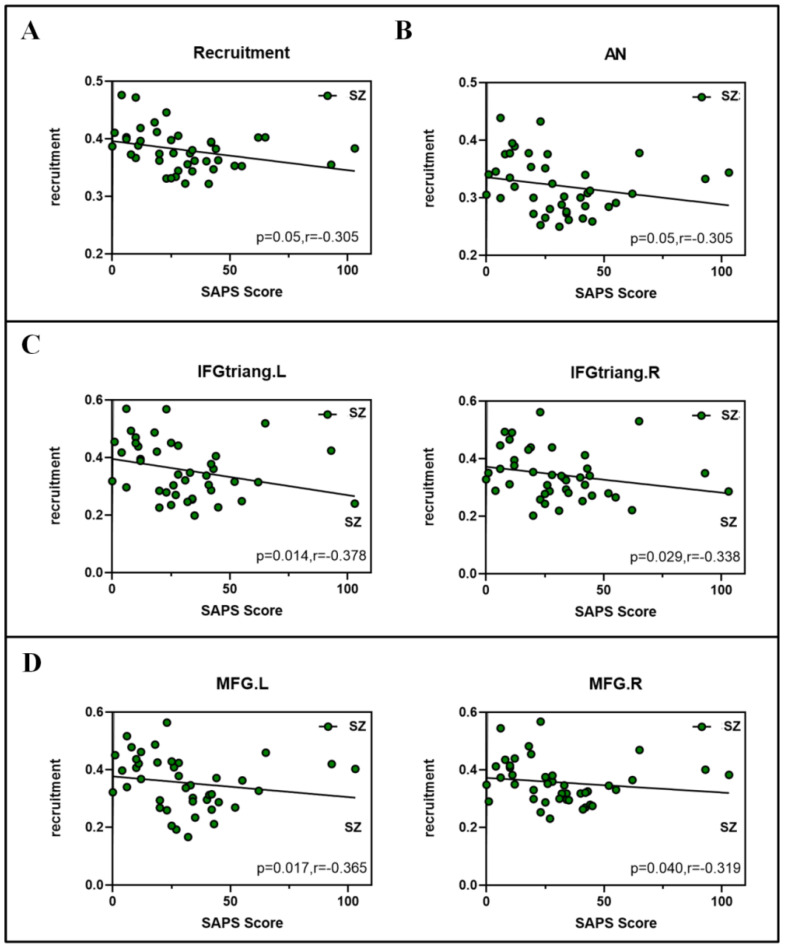
Spearman correlations between dynamic properties and ASRS scores in slow3 (0.073–0.198 Hz). (**A**) Recruitment (**B**) Recruitment in the attention network (AN). (**C**) Recruitment in the inferior frontal gyrus, triangular part (IFGtriang) and (**D**) middle front gyrus (MFG).

**Table 1 brainsci-12-00727-t001:** Demographic and clinical characteristics.

Characteristic	SZ	NC	Statistical Test
Number of subjects	42	40	--
Age (years)	35.19 ± 8.37	32.25 ± 8.81	P = 0.125
Sex (male/female)	30/12	25/15	P = 0.396
SAPS	30.67 ± 22.26	--	--

Note: The values are denoted as mean ± standard deviation.

**Table 2 brainsci-12-00727-t002:** The number of modules in full frequency (0.01–0.25 Hz), slow5 (0.01–0.027 Hz), slow4 (0.027–0.073 Hz), slow3 (0.073–0.198 Hz), and slow2 (0.198–0.25 Hz).

NC					SZ				
0.01–0.25 Hz	Slow5	Slow4	Slow3	Slow2	0.01–0.25 Hz	Slow5	Slow4	Slow3	Slow2
5.2	6.4	6	7.4	4.6	5.8	7	4.8	5.4	4.6
7.6	5.4	7	6.6	7.2	6.6	6	6.8	7.6	5.4
6.6	6.2	6.2	5.8	6.2	5.2	7.6	6.2	6.4	4.8
6	8.6	6.2	5.4	5.2	6.4	7	7	5.6	6
7	7.8	6.8	7.6	6.2	6.2	6.2	6	6.4	4.6
6.6	5	4.6	6.8	5.2	6	6	6	6.8	4.4
7	7.8	7.4	7.8	6.8	6	6	5.2	7.8	5.8
5.8	4.8	6.4	8.2	6.8	5.2	7.6	7.4	5	4.8
5.8	7.4	6	7	5.8	5.6	5.4	5.4	6.2	7
7.4	7.6	7.4	6.4	6.4	7.2	7	6.4	7.8	6.8
6.2	6.8	7	7.2	5.4	6.6	4.2	6.4	6.2	4.8
6	7	5.4	6.6	4.8	5.2	6.4	3.6	6.8	5
7.4	6	7.2	8.2	5.6	6	5.8	5.2	5.8	6
7.4	7.4	7	5.8	7.8	6.4	5.2	6.2	7.6	5.6
6.8	6.2	8.2	6.2	5.8	7.4	6.6	7.2	9.4	7
7.2	6	7.2	4.8	6.4	6.4	6.4	7	7.2	5
7.4	7.2	6.8	7.2	7.6	8	5.6	7	9	4.8
7.8	8.6	7.6	5.4	7	6.6	6	5.6	5.4	7
6.4	6.6	8	6.2	6.2	6.4	6.8	6.8	7.6	5.2
6.2	6.8	6.2	7.8	5.2	7.2	8.6	6.8	7	4.2
6	5.8	5.6	6.4	6.8	7.6	5.2	7.2	6.4	6.4
7	5.8	5.2	7.4	6.2	5.6	5.2	4	6.4	5.4
7	5.2	5	4.8	5.2	6.2	6.2	7.4	8.2	6.2
5.8	7.4	7.2	5	7.6	7.8	6.4	6.6	6	6.2
6.2	7	5.2	7.6	7.2	5	6.6	5.4	5.8	6.6
5.4	5.8	5.4	7.4	6.2	6.6	7.6	6.8	8	4.8
7.8	7.2	7.4	7.4	4.4	7.8	7.2	6.8	6.2	4.6
6.8	6.8	6.4	7.6	6.8	6	5.8	5.6	7.4	6.8
7.6	6.6	7	5.8	5.4	6.2	5.2	5.6	6	6.8
5.6	6.6	8.2	6.6	4.8	6.8	7	7	6.4	6.6
6.6	6.4	7.2	7.4	7.4	5.6	6.4	4.6	9.2	4.6
5.6	7.6	5.4	6.6	6.2	6.2	7	8	6.4	4.6
7	6.8	5.4	6.2	5.2	7	7.4	5.2	5.4	5.2
6.2	7.6	6.6	7.2	6.4	6.2	7.2	6.4	7.2	6.8
7	7.8	7.4	7.6	7.2	6.2	7.4	7.6	5.4	5.4
6.4	6.2	4.8	5.2	5	6.6	7.4	7.2	7.6	6
6.6	9.6	7.8	8.4	6	7.4	7.4	6.8	6.8	4.6
7.2	8.6	8	7	5.6	6.8	6.8	7	7.8	4
6.8	7.2	8.2	6.6	6	6.8	6.6	7.4	7.8	4.6
5.6	7.8	6	5.8	7	6.8	8.2	6.4	7	4.4
\	\	\	\	\	6.2	6.2	5	8.4	6.6
\	\	\	\	\	8	6.8	8	8.2	5.6

**Table 3 brainsci-12-00727-t003:** Integration and Recruitment of brain networks in full frequency (0.01–0.25 Hz), slow5 (0.01–0.027 Hz), slow4 (0.027–0.073 Hz), slow3 (0.073–0.198 Hz), and slow2 (0.198–0.25 Hz).

Characteristic		0.01–0.25 Hz	Slow5	Slow4	Slow3	Slow2
Integration	SMN	T = −1.277 P = 0.205	T = 0.478 P = 0.634	T = 0.171 P = 0.865	T = −0.246 P = 0.807	T = 0.073 P = 0.942
	VN	T = 1.012 P = 0.314	T = 0.279 P = 0.781	T = 2.263 P = 0.026 *	T = 0.522 P = 0.603	T = 0.522 P = 0.603
	AN	T = −0.781 P = 0.437	T = 0.639 P = 0.525	T = 0.378 P = 0.707	T = 0.109 P = 0.914	T = 1.124 P = 0.264
	DMN	T = −1.010 P = 0.316	T = 0.455 P = 0.651	T = 0.048 P = 0.962	T = −0.564 P = 0.574	T = 1.394 P = 0.167
	LSN	T = −1.610 P = 0.111	T = −0.041 P = 0.967	T = 1.457 P = 0.149	T = −0.576 P = 0.566	T = 1.031 P = 0.305
Recruitment	SMN	T = −0.783 P = 0.436	T = 1.761 P = 0.082	T = −0.677 P = 0.501	T = −0.196 P = 0.845	T = 0.889 P = 0.377
	VN	T = −2.840 P = 0.006 **	T = −0.701 P = 0.485	T = −4.101 P = 0.000 ***	T = −0.830 P = 0.409	T = −0.272 P = 0.787
	AN	T = −2.392 P = 0.019 *	T = 0.07 P = 0.945	T = −3.557 P = 0.001 ***	T = −1.266 P = 0.209	T = 0.195 P = 0.846
	DMN	T = −1.123 P = 0.265	T = 1.352 P = 0.180	T = −2.178 P = 0.032 *	T = 0.390 P = 0.698	T = −0.803 P = 0.424
	LSN	T = −0.713 P = 0.478	T = 1.489 P = 0.140	T = 2.456 P = 0.016 *	T = −0.431 P = 0.668	T = 0.163 P = 0.871

Asterisk indicates pairwise group differences; *** represents *p* < 0.001, ** represents *p* < 0.01, * represents *p* < 0.05.

**Table 4 brainsci-12-00727-t004:** RSN-to-RSN integration scores with significant differences in slow3 (0.073–0.198 Hz).

RSN1	RSN2	NC (SD)	SZ (SD)	P (FDR)
Visual network (VN)	Attention network (AN)	0.194 (0.055)	0.235 (0.055)	0.000
Visual network (VN)	Default mode network (DMN)	0.212 (0.049)	0.240 (0.036)	0.001

**Table 5 brainsci-12-00727-t005:** RSN-to-RSN integration scores in full frequency (0.01–0.25 Hz), slow5 (0.01–0.027 Hz), slow4 (0.027–0.073 Hz), slow3 (0.073–0.198 Hz), and slow2 (0.198–0.25 Hz).

Frequency	RSN1	RSN2	NC (SD)	SZ (SD)	P (FDR)
0.01–0.25 Hz	visual network (VN)	somatosenery/motor and auditory network (SMN)	0.258 (0.115)	0.267 (0.132)	0.731
	visual network (VN)	attention network (AN)	0.168 (0.082)	0.183 (0.062)	0.367
	visual network (VN)	default mode network (DMN)	0.201 (0.071)	0.213 (0.055)	0.774
	visual network (VN)	limbic/paralimbic and subcortical network (LSN)	0.165 (0.069)	0.180 (0.056)	0.312
Slow5	visual network (VN)	somatosenery/motor and auditory network (SMN)	0.348 (0.110)	0.344 (0.126)	0.858
	visual network (VN)	attention network (AN)	0.292 (0.088)	0.295 (0.069)	0.839
	visual network (VN)	default mode network (DMN)	0.321 (0.078)	0.329 (0.068)	0.639
	visual network (VN)	limbic/paralimbic and subcortical network (LSN)	0.294 (0.067)	0.307 (0.089)	0.469
Slow4	visual network (VN)	somatosenery/motor and auditory network (SMN)	0.311 (0.119)	0.299 (0.108)	0.613
	visual network (VN)	attention network (AN)	0.210 (0.075)	0.215 (0.071)	0.706
	visual network (VN)	default mode network (DMN)	0.254 (0.069)	0.260 (0.069)	0.666
	visual network (VN)	limbic/paralimbic and subcortical network (LSN)	0.232 (0068)	0.236 (0.071)	0.774
Slow3	visual network (VN)	somatosenery/motor and auditory network (SMN)	0.271 (0.074)	0.271 (0.071)	0.552
	visual network (VN)	attention network (AN)	0.194 (0.055)	0.235 (0.055)	0.000
	visual network (VN)	default mode network (DMN)	0.212 (0.049)	0.240 (0.036)	0.001
	visual network (VN)	limbic/paralimbic and subcortical network (LSN)	0.220 (0.052)	0.215 (0.043)	0.797
Slow2	visual network (VN)	somatosenery/motor and auditory network (SMN)	0.313 (0.055)	0.311 (0.061)	0.856
	visual network (VN)	attention network (AN)	0.294 (0.035)	0.297 (0.054)	0.759
	visual network (VN)	default mode network (DMN)	0.284 (0.041)	0.290 (0.044)	0.458
	visual network (VN)	limbic/paralimbic and subcortical network (LSN)	0.269 (0.042)	0.280 (0.051)	0.271

**Table 6 brainsci-12-00727-t006:** Brain map of regions with significant differences on integration in slow3 (0.073–0.198 Hz).

Frequency	Network	Name	Abb	ROI	NC (SD)	SZ (SD)	P (FDR)
Slow3	SMN	Rolandic_Oper	ROL.L	17	0.289 (0.053)	0.245 (0.053)	0.001
			ROL.R	18	0.279 (0.041)	0.243 (0.053)	0.011
		Insula	INS.L	29	0.300 (0.055)	0.258 (0.062)	0.015
			INS.R	30	0.300 (0.055)	0.260 (0.060)	0.015
		Heschl	HES.L	79	0.283 (0.049)	0.244 (0.052)	0.011
			HES.R	80	0.280 (0.044)	0.244 (0.048)	0.011
		Temporal_Sup	STG.L	81	0.286 (0.051)	0.253 (0.055)	0.036
			STG.R	82	0.284 (0.049)	0.243 (0.047)	0.000
	VN	Lingual	LING.L	47	0.212 (0.053)	0.248 (0.050)	0.015
			LING.R	48	0.208 (0.052)	0.239 (0.054)	0.045
		Fusiform_R	FFG.R	56	0.241 (0.058)	0.288 (0.047)	0.000
	DMN	Cingulum_Ant	ACG.L	31	0.297 (0.052)	0.260 (0.052)	0.015
			ACG.R	32	0.297 (0.054)	0.263 (0.052)	0.032
	LSN	Amygdala_R	AMYG.R	42	0.272 (0.056)	0.233 (0.046)	0.011
		Temporal_Pole_Mid	TPOmid.L	87	0.280 (0.052)	0.248 (0.046)	0.028
			TPOmid.R	88	0.294 (0.058)	0.247 (0.041)	0.000

**Table 7 brainsci-12-00727-t007:** Brain map of regions with significant differences on recruitment in full frequency (0.01–0.25 Hz) and slow3 (0.073–0.198 Hz).

Frequency	Network	Name	Abb	ROI	NC (SD)	SZ (SD)	P (FDR)
0.01–0.25 Hz	VN	Lingual_L	LING.L	47	0.743 (0.143)	0.634 (0.169)	0.037
		Occipital_Sup	SOG.L	49	0.769 (0.114)	0.676 (0.145)	0.037
			SOG.R	50	0.769 (0.118)	0.661 (0.158)	0.037
		Fusiform_R	FFG.R	56	0.534 (0.247)	0.371 (0.219)	0.037
	LSN	Cingulum_Mid	DCG.L	33	0.287 (0.131)	0.200 (0.097)	0.037
			DCG.R	34	0.281 (0.122)	0.199 (0.106)	0.037
Slow3	SMN	Insula_L	INS.L	29	0.456 (0.116)	0.395 (0.110)	0.048
	VN	Lingual_L	LING.L	47	0.610 (0.129)	0.477 (0.143)	0.001
		Occipital_Inf_L	IOG.L	53	0.494 (0.157)	0.405 (1.173)	0.048
		Fusiform	FFG.L	55	0.312 (0.166)	0.232 (0.112)	0.040
			FFG.L	56	0.349 (0.176)	0.233 (0.115)	0.004
	AN	Frontal_Mid	MFG.L	7	0.421 (0.960)	0.355 (0.093)	0.010
			MFG.R	8	0.429 (0.088)	0.357 (0.075)	0.002
		Frontal_Inf_Oper	IFGoperc.L	11	0.384 (0.112)	0.328 (0.093)	0.048
			IFGoperc.R	12	0.375 (0.107)	0.317 (0.086)	0.031
		Frontal_Inf_Tri	IFGtriang.L	13	0.434 (0.091)	0.357 (0.098)	0.003
			IFGtriang.R	14	0.423 (0.095)	0.345 (0.086)	0.002
		Frontal_Inf_Orb	ORBinf.L	15	0.394 (0.085)	0.346 (0.088)	0.048
			ORBinf.R	16	0.399 (0.101)	0.334 (0.082)	0.011
		Parietal_Inf	IPL.L	61	0.365 (0.101)	0.297 (0.068)	0.004
			IPL.R	62	0.369 (0.106)	0.317 (0.067)	0.034
		Angular	ANG.L	65	0.365 (0.101)	0.313 (0.074)	0.034
			ANG.R	66	0.372 (0.099)	0.316 (0.082)	0.023
		Temporal_Inf_L	ITG.L	89	0.309 (0.093)	0.260 (0.089)	0.048
	DMN	Frontal_Sup_Medial	SFGmed.L	23	0.442 (0.096)	0.385 (0.078)	0.017
			SFGmed.R	24	0.446 (0.091)	0.387 (0.081)	0.011
		Cingulum_Post_L	PCG.L	35	0.340 (0.111)	0.287 (0.081)	0.048
		Temporal_Mid	MTG.L	85	0.341 (0.108)	0.268 (0.076)	0.004
			MTG.R	86	0.324 (0.112)	0.258 (0.072)	0.010
	LSN	Cingulum_Mid	DCG.L	33	0.270 (0.107)	0.184 (0.074)	0.001
			DCG.R	34	0.267 (0.101)	0.189 (0.073)	0.002
		Hippocampus	HIP.L	37	0.396 (0.075)	0.479 (0.099)	0.001
			HIP.R	38	0.389 (0.080)	0.483 (0.105)	0.001
		ParaHippocampal	PHG.L	39	0.395 (0.071)	0.468 (0.103)	0.003
			PHG.R	40	0.389 (0.074)	0.470 (0.106)	0.002
		Amygdala	AMYG.L	41	0.392 (0.082)	0.466 (0.108)	0.005
			AMYG.R	42	0.386 (0.089)	0.469 (0.108)	0.003
		Temporal_Pole_Mid	TPOmid.L	87	0.324 (0.085)	0.423 (0.131)	0.002
			TPOmid.R	88	0.336 (0.099)	0.409 (0.123)	0.017

## Data Availability

https://openfmri.org/dataset/ds000030/ (accessed on 23 March 2022).

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
