# Peer review of "Frequency-Specific Analysis of the Dynamic Reconfiguration of the Brain in Patients with Schizophrenia"

_brainsci, 2022, doi:10.3390/brainsci12060727_

Round 1
Reviewer 1 Report
In their manuscript entitled ‘Frequency-specific analysis of the dynamic reconfiguration of the brain in patients with schizophrenia’ the authors show that the alterations in dynamic functional connectivity in schizophrenia are frequency specificity, exhibiting the most significant differences with respect to healthy controls in the 0.073-0.198Hz frequency range. Moreover, the authors demonstrate that only some specific resting-state networks are affected in schizophrenia, and these relate to specific scores of the disease.
This is a relevant and timely subject, given the growing effort to identify evermore precise markers of psychiatric disorders in neuroimaging data, which may not only provide better insights into the pathophysiology of these disorders but also assist in their accurate diagnosis.
Although the study is well designed and the results are solid and meaningful, the manuscript needs a thorough and careful revision of the writing (both in terms of English but also in terms of clarity) before publication in Brain Sciences. This is really critical, otherwise the article is very hard to read. Improving the clarity of this work will certainly improve its impact in the field.
Comments:
Abstract
1) ‘The dynamic reconfiguration of brain community in schizophrenia usually focuses on low-frequency (0.01-0.08Hz), but it cannot completely cover the complex neural activity patterns in the resting-state brain.’
Please revise this phrase. Although I can understand what the authors mean, the sentence does not make sense. For instance, the subject for the verbs ‘focuses’ and ‘cannot’ is ‘reconfiguration’. (‘The [..] reconfiguration […] focuses on […], but it cannot […]. ) Please revise.
2) ‘Studies have shown that each specific frequency band can describe the unique spontaneous fluctuations of neural activities in the brain. The high-frequency band can also provide valuable information for the diagnosis of schizophrenia.’
Please report the frequency range corresponding to ‘high frequency’.
Also, please revise the sentences. Suggestion: ‘Studies have shown that distinct frequency bands can capture unique fluctuations in brain activity, with high-frequency components (X-Y Hz) providing valuable information for the diagnosis of schizophrenia.’
3) (full frequency, slow2 to slow5).
the authors refer to ‘slow2’ and ‘slow5’ without introducing it. In the abstract, it is sufficient to report the frequency ranges, without the need to use the terms ‘slow2 or slow5’.
4) ‘In addition, visual network, attention network, and default mode network show high differences and connectivity, indicating that the abnormal state of the brain in these regions will affect the functional network of schizophrenia.’
This sentence also needs revision. ‘high differences and connectivity’? ‘abnormal state of the brain in these regions’?
Introduction
5) ‘ Functional magnetic resonance imaging (fMRI) studies are generally based on static brain networks for analysis, but fluctuations in brain networks are usually observed in dynamic functional connectivity (FC) analysis[5].’
Again, this is confusing. Most fMRI studies do not even look at networks (i.e., task-based activations of the hemodynamic response function)’. Please clarify that network analysis generally refer to resting-state studies.
6) ‘ For example, George used a modeling method called the dynamic modular organization to in-35 vestigate better the inter-group differences in dynamic community structure in SZ[6].
It has to be ‘Gifford and colleagues’ instead of ‘George’. (same for the other references in the text, when the article has more than one author, please include ‘and colleagues’.
7) There is a recent paper on dynamic functional connectivity in schizophrenia that the authors should refer to in the introduction.
Farinha, M., Amado, C., Morgado, P. and Cabral, J., 2022. Increased excursions to functional networks in schizophrenia in the absence of task. Frontiers in Neuroscience, 16. https://doi.org/10.3389/fnins.2022.821179
Methods
8) ‘five frequency bands are divided here: full frequency(0.01-0.25Hz), slow2 (0.198– _0.25 Hz), slow3 (0.073–0.198 Hz), slow4 (0.027–0.073 Hz) and slow5 (0.01– 027 Hz);
It is not common to sort the frequencies from the faster to the slower. I would suggest sorting them from slower to faster, but this is a decision of the authors.
Results
9) Please report the frequency ranges in figures 2, 3 and 4 to facilitate interpretation, otherwise the reader needs to constantly go back to the methods section to check which are the corresponding bands. The same applies throughout the text.
Also, use adequate 'font size' and the same 'font type' in the figures.
Discussion
10) ‘NF ho[48] also believed that amygdala orbito- frontal functional connectivity decreased in schizophrenic patients in the study of resting functional connectivity.’
I suggest replacing the verb ‘believe’ (which does not have any scientific validity) by ‘demonstrated’ or ‘showed’. Also, please use ‘Ho and colleagues’ as explained before.
Conclusion
11) ‘indicating that high-frequency fMRI signals could change spontaneous neural activity,’
This sentence is making a claim that is not verified. The role of high-frequency fMRI signals is unclear and, from what I understood of this work, it has not been shown in any way that the fMRI signals can ‘change’ spontaneous neural activity.
Author Response
Response to Reviewer 1 Comments
Abstract
- Point 1: Page 1 of 23, please revise this phrase. “The dynamic reconfiguration of brain community in schizophrenia usually focuses on low-frequency (0.01-0.08Hz), but it cannot completely cover the complex neural activity patterns in the resting-state brain. ”
Response 1: Thank you very much for pointing this out. I am so sorry we did not describe it clearly. We have changed the description in its abstract in the revised manuscript and highlighted the fixed part in gray.
- Point 2: Page 1 of 23, also, please revise the sentences. “Studies have shown that each specific frequency band can describe the unique spontaneous fluctuations of neural activities in the brain. The high-frequency band can also provide valuable information for the diagnosis of schizophrenia. ”
Response 2: Thank you very much for your constructive suggestions. We revised the interpretation in the abstract and highlighted these changes in gray. In addition, concerning frequency, we can only specify that the low-frequency is 0.01-0.1Hz. Up to now, there is no clear range of high-frequency. Here, the frequency bands greater than 0.1Hz are collectively referred to as high-frequency. In the study of [1], higher than 0.1Hz is also considered as high-frequency.
- Point 3: Page 1 of 23, the authors refer to ‘slow2’ and ‘slow5’ without introducing it. In the abstract, it is sufficient to report the frequency ranges, without the need to use the terms ‘slow2 or slow5’.
Response 3: Thank you very much for your constructive suggestions. We have revised this part in abstract and marked the changes with gray highlights.
- Point 4: Page 1 of 23, this sentence also needs revision. ‘high differences and connectivity’? ‘abnormal state of the brain in these regions’?
Response 4: Thank you very much for your good suggestions. We have revised the abstract section accordingly and marked the changes with gray highlights.
Introduction
- Point 5: Page 1 of 23, again, this is confusing. Most fMRI studies do not even look at networks (i.e., task-based activations of the hemodynamic response function)’. Please clarify that network analysis generally refer to resting-state studies.
Response 5: Thank you very much for pointing this out. We are sorry that we made some mistakes in the introduction of the functional magnetic resonance imaging (fMRI) studies. In functional magnetic resonance imaging (fMRI) studies, traditional brain network construction methods usually during the resting-state (i.e., a static network)[2]. Current dynamic network analyses have confirmed that fluctuations in functional connections exist, which has attracted increasing attention in the academic world [3, 4]. We have revised the explanations in page 1 of 19 and marked the changes with gray highlights.
- Point 6: Page 1 of 23, it has to be ‘Gifford and colleagues’ instead of ‘George’. (same for the other references in the text, when the article has more than one author, please include ‘and colleagues’.
Response 6: Thank you very much for your constructive suggestions. We have revised the explanations in page 1 of 19 and marked the changes with gray highlights. At the same time, we have also made corresponding modifications to this issue in other parts.
- Point 7: Page 1 of 23, there is a recent paper on dynamic functional connectivity in schizophrenia that the authors should refer to in the introduction.
Response 7: Thank you very much for your good suggestions. This reference is beneficial to us. We have referred to in the introduction in section 1 with highlighting in gray.
Methods
- Point 8: Page 3 of 23, it is not common to sort the frequencies from the faster to the slower. I would suggest sorting them from slower to faster, but this is a decision of the authors.
Response 8: Thank you very much for pointing this out. We have sorted the frequency from slower to faster and highlighted the revised part in gray.
Results
- Point 9: Page 6 of 23, please report the frequency ranges in figures 2, 3 and 4 to facilitate interpretation, otherwise the reader needs to constantly go back to the methods section to check which are the corresponding bands. The same applies throughout the text. Also, use adequate 'font size' and the same 'font type' in the figures.
Response 9: Thank you very much for pointing this out. We have reported the frequency ranges in all figures to facilitate interpretation.
Discussion
- Point 10: Page 11of 23, I suggest replacing the verb ‘believe’ (which does not have any scientific validity) by ‘demonstrated’ or ‘showed’. Also, please use ‘Ho and colleagues’ as explained before.
Response 10: Thank you very much for your constructive suggestions. I am so sorry we did not describe it clearly. We have corrected the corresponding errors and highlighted the revised part in gray.
Conclusion
- Point 11: Page 12 of 23, this sentence is making a claim that is not verified. The role of high-frequency fMRI signals is unclear and, from what I understood of this work, it has not been shown in any way that the fMRI signals can ‘change’ spontaneous neural activity.
Response 11: Thank you very much for your good suggestions. I am so sorry we did not make a clear description. We have deleted this inappropriate remark in conclusion. Thank you again for your suggestions.
References
- Yu, X., et al., Frequency-specific abnormalities in regional homogeneity among children with attention deficit hyperactivity disorder: a resting-state f MRI study. 2016.
- Cui, X., et al., Analysis of Dynamic Network Reconfiguration in Adults with Attention-Deficit/Hyperactivity Disorder Based Multilayer Network. 2021.
- Allen, E.A., et al., Tracking Whole-Brain Connectivity Dynamics in the Resting State. 2014(3): p. 663-676.
- Farinha, M., et al., Increased excursions to functional networks in schizophrenia in the absence of task. 2022. 16.

Reviewer 2 Report
In "Frequency-specific analysis of the dynamic reconfiguration of the brain in patients with schizophrenia ", Yang et al. explored dynamic re-configurability of brain network structure in various frequency bands for schizophrenia (SZ) patients. They used multilayer community detection to identify brain community of each participant. Recruitment and integration obtained from the module allegiance matrix were the two main metrics for various analyses. This is a well conducted and well reported study but there are certain aspects that need clarification and/or further analysis for the study to be in publishable form.
Specific comments to the authors are given below.
Section 1:
- The authors should provide reference from the literature for breakdown of the frequencies in various bands.
Section 2:
- Based on the study referred to in the manuscript (Leonardi and Van De Ville 2015), the window length should be 1/fmin , where fmin is the minimum frequency in the signals under consideration. Two other related studies (Shakil S, Keilholz SD et al. 2015, Shakil S, Billings JCW et al. 2017) also explored the frequency dependence of the window length and showed that this length is dependent on the ratio of minimum and maximum frequencies in the correlating signals. Based on these studies, the window length for each frequency band should be different based on band’s own minimum frequency (and ratios of the frequencies), which is not the case in this study. The authors should comment how this may influence the results and what would happen if the window length is different for each band?
- What is the structure of the matrices Hkf? Are they diagonal matrices or have off-diagonal non-zero values, too?
- Is there any relationship between Pijt in Section 2.4 and Pij in Section 2.5? If not, then it would be better that authors state it somewhere to avoid any confusion in readers’ minds.
- Is the T in Pij formula representing number of slices or number of layers (windowed correlation matrices)? The word layers and slice both are used in Section 2.5.1.
Section 3:
- Is the Figure 2. showing results of one subject or average over all the subjects?
- Why only full frequency and slow 3 are used to analyze RSN to RSN integration comparison in Section 3.3.
Section 4:
- How does the reduction in recruitment influence the cognitive ability in SZ patients in Section 4.1?
- The results and discussion show that the most significant band showing differences in SZ and NC is slow 3. Are there any studies in the literature reporting significance of this band for the same purpose? How do these results compare with any similar studies done in the past?
- In Section 4.3, the authors report that the reason for slow 3 band showing significant results maybe since it contains the frequency range (0.073 – 0.1 Hz) mostly studied in rs-fMRI study. Can this mean that the results are significant due to presence of these frequencies and not because of the other frequencies (0.1 – 0.198 Hz) in slow 3? I suggest that the authors perform the analysis by breaking this band into two sub-bands of frequencies ranges (0.073 -0.1 Hz and 0.1 – 0.198 Hz). It is important to do this since if the results are significant in sub-band of 0.073 -0.1 Hz only then it may mean that the standard method of working with this frequency range should be enough and there is no need to analyze various frequency bands in case of resting-state fMRI studies.
References:
Leonardi, N. and D. Van De Ville (2015). "On spurious and real fluctuations of dynamic functional connectivity during rest." Neuroimage 104: 430-436.
Shakil S, Billings JCW, Keilholz S and Lee C.-H (2017). "Parametric Dependencies of Sliding Window Correlation." IEEE Transactions on Biomedical Engineering 65(2): 254-263.
Shakil S, Keilholz SD and Lee C.-H (2015). On Frequency Dependencies of Sliding Window Correlation IEEE conference on BioInformatics and BioEngineering. Maryland.
Author Response
Response to Reviewer 2 Comments
Section 1:
- Page 3 of 23, the authors should provide reference from the literature for breakdown of the frequencies in various bands.
Response 1:Thank you very much for your good suggestions. I'm so sorry we didn't explain this part clearly. We have added the references in Section 2.2 and a note in blue on page 3 of 23.
Section 2:
- Page 4 of 23, based on studies, the window length for each frequency band should be different based on band’s own minimum frequency (and ratios of the frequencies), which is not the case in this study. The authors should comment how this may influence the results and what would happen if the window length is different for each band?
Response 1: Thank you very much for your constructive suggestions. I am so sorry we did not describe it clearly. According to recommendations of previous research, the window size used in dynamic network studies should be no less than , where is the minimum frequency of data[1, 2]. Therefore, the length of the time window is 1/0.01Hz = 100s (50 time points).However, we ignore the frequency dependence of the window length[3, 4].We have added the recruitment and integration of each frequency band under different window lengths in the supplementary materials, and added a note in blue on page 4 of 23. The specific correspondence figures as follows:
Figure 1. Window length = 20; The difference of recruitment and integration from 0.01-0.25Hz and slow5、slow4、slow3、slow2 in the whole brain level between NC and SZ. (A) Integration. (B) Recruitment. Asterisk indicates pairwise group differences; * represents p < 0.05; ** represents p < 0.01.
Figure 2. Window length = 30; The difference of recruitment and integration from 0.01-0.25Hz and slow5、slow4、slow3、slow2 in the whole brain level between NC and SZ. (A) Integration. (B) Recruitment. Asterisk indicates pairwise group differences; * represents p < 0.05; ** represents p < 0.01.
Figure 3. Window length = 40; The difference of recruitment and integration from 0.01-0.25Hz and slow5、slow4、slow3、slow2 in the whole brain level between NC and SZ. (A) Integration. (B) Recruitment. Asterisk indicates pairwise group differences; * represents p < 0.05; ** represents p < 0.01.
According to the experimental conclusion, we found that the choice of different window lengths had no significant effect on the results. Here, we select the minimum value of the whole frequency band (0.01Hz) as the frequency for dividing the time window.
- Page 4 of 23, what is the structure of the matrices ? Are they diagonal matrices or have off-diagonal non-zero values, too ?
Response 2: Thank you very much for your constructive suggestions. I am so sorry we did not describe it clearly. After reading some literature, the research on multilayer brain networks only retains the edges of the same nodes between layers. On this basis, the matrices is a diagonal matrices. We connected nodes in one time window to themselves in adjacent time windows to represent time dependence. A simple representation of multilayer networks is through an adjacency tensor called a supra-adjacency matrix[5, 6], which is a square matrix of (N is the number of nodes in one window). is the connection matrix between layer and layer . Specifically, matrices and values are the same. In the revised manuscript, we have added an explanation of “the matrices ” in the blue font on page 4 of 23.
- Page 4 of 23, is there any relationship between in Section 2.4 and in Section 2.5? If not ,then it would be better that authors states it somewhere to avoid any confusion in readers’ minds.
Response 3: Thank you very much for your precise suggestions. I am so sorry we did not notice this problem because of my negligence. There is no relationship between in Section 2.4 and in Section 2.5. The element in Section 2.4 represents the component of layer matrix corresponding to the optimized zero models. The element in Section 2.5 represents the probability that node and node are assigned to the same community during the whole scanning process. We have revised the in Section 2.4 to in the blue font on page 4 of page 23.
- Page 5 of 23, is the in formula representing number of slices or number of layers (windowed correlation matrices)? The word layers and slice both are used in Section 2.5.1.
Response 4: Thank you very much for your good suggestions. The in formula representing number layers. We have revised the discussion section accordingly and marked the changes with gray highlights.
Section 3:
- Page 6 of 23, is the Figure 2. Showing results of one subject or average over all the subjects?
Response 1: Thank you very much for your question. Here, we studied 42 schizophrenia (SZ) and 40 normal controls (NC). The results in Figure 2 show the average values of all subjects.
- Page 6 of 23, why only full frequency and slow3 are used to analyze RSN to RSN integration comparison in Section 3.3.
Response 2: Thank you very much for your question. Based on the results of the RSN level, we found a difference between the full frequency (0.01-0.25Hz) and slow3 (0.073-0.198Hz). To better understand the integration relationship between RSNs, we studied the slow3 of different modules. At the same time, we have added the results between RSN to RSN of other frequency bands (slow5、slow4、slow2) in the supplementary material and added a note in blue on page 17 of 19. The results show no significant difference between RSN to RSN in different frequency bands. The specific correspondence table is as follows:
Table 6. RSN-to-RSN integration scores in 0.01-0.25Hz、slow5、slow4、slow3、slow2. |
|||||
Frequency |
RSN1 |
RSN2 |
NC(SD) |
SZ(SD) |
P(FDR) |
0.01-0.25Hz |
visual network(VN) |
somatosenery/motor and auditory network (SMN) |
0.258(0.115) |
0.267(0.132) |
0.731 |
visual network(VN) |
attention network(AN) |
0.168(0.082) |
0.183(0.062) |
0.367 |
|
visual network(VN) |
default mode network(DMN) |
0.201(0.071) |
0.213(0.055) |
0.774 |
|
visual network(VN) |
limbic/paralimbic and subcortical network(LSN) |
0.165(0.069) |
0.180(0.056) |
0.312 |
|
Slow5 |
visual network(VN) |
somatosenery/motor and auditory network (SMN) |
0.348(0.110) |
0.344(0.126) |
0.858 |
visual network(VN) |
attention network(AN) |
0.292(0.088) |
0.295(0.069) |
0.839 |
|
visual network(VN) |
default mode network(DMN) |
0.321(0.078) |
0.329(0.068) |
0.639 |
|
visual network(VN) |
limbic/paralimbic and subcortical network(LSN) |
0.294(0.067) |
0.307(0.089) |
0.469 |
|
Slow4 |
visual network(VN) |
somatosenery/motor and auditory network (SMN) |
0.311(0.119) |
0.299(0.108) |
0.613 |
visual network(VN) |
attention network(AN) |
0.210(0.075) |
0.215(0.071) |
0.706 |
|
visual network(VN) |
default mode network(DMN) |
0.254(0.069) |
0.260(0.069) |
0.666 |
|
visual network(VN) |
limbic/paralimbic and subcortical network(LSN) |
0.232(0068) |
0.236(0.071) |
0.774 |
|
Slow3 |
visual network(VN) |
somatosenery/motor and auditory network (SMN) |
0.271(0.074) |
0.271(0.071) |
0.552 |
visual network(VN) |
attention network(AN) |
0.194(0.055) |
0.235(0.055) |
0.000 |
|
visual network(VN) |
default mode network(DMN) |
0.212(0.049) |
0.240(0.036) |
0.001 |
|
visual network(VN) |
limbic/paralimbic and subcortical network(LSN) |
0.220(0.052) |
0.215(0.043) |
0.797 |
|
Slow2 |
visual network(VN) |
somatosenery/motor and auditory network (SMN) |
0.313(0.055) |
0.311(0.061) |
0.856 |
visual network(VN) |
attention network(AN) |
0.294(0.035) |
0.297(0.054) |
0.759 |
|
visual network(VN) |
default mode network(DMN) |
0.284(0.041) |
0.290(0.044) |
0.458 |
|
visual network(VN) |
limbic/paralimbic and subcortical network(LSN) |
0.269(0.042) |
0.280(0.051) |
0.271 |
Section 4:
- Page 10 of 23, how does the reduction in recruitment influence the cognitive ability in SZ patients in Section 4.1?
Response 1: Thank you very much for your question. We can explain this problem through the Correlation between network measures and SAPS scores. In Figure 7, we found a negative correlation between the average recruitment score and the SAPS score at the whole-brain level. The lower recruitment, the higher SAPA. SAPS is mainly used to evaluate the positive symptoms of schizophrenia, including hallucinations, delusions, and other symptoms. The higher the SAPA score, the more serious the SZ illness and the worse the SZ cognitive ability. Therefore, here, we believe that the decline in recruitment will affect the cognitive ability of SZ. We have changed the description in the revised manuscript and highlighted the fixed part in gray.
- Page 11 of 23, the results and discussion show that the most significant band showing differences in SZ and NC is slow 3. Are there any studies in the literature reporting significance of this band for the same purpose? How do these results compare with any similar studies done in the past?
Response 2: Thank you very much for your good suggestions. I am so sorry we did not make a clear description—some studies in the literature report some conclusions about this band (slow3:0.073-0.198Hz).For example, Wang and colleagues [7]obtained higher accuracy in slow3 by using hierarchical sparse learning method to diagnose schizophrenia (SZ) in resting-state functional magnetic resonance imaging (rs-fMRI). Zuo and colleagues[8] analyzed the classification weight distribution and concluded that slow3 had the best classification performance while slow2 had the worst.
All the above studies show that slow3 is of research value. We also got similar results. From the perspective of dynamic reconfiguration of the brain network, the recruitment and integration are significant in the slow3.
- In Section 4.3, the authors report that the reason for slow 3 band showing significant results maybe since it contains the frequency range (0.073 – 0.1 Hz) mostly studied in rs-fMRI study. Can this mean that the results are significant due to presence of these frequencies and not because of the other frequencies (0.1 – 0.198 Hz) in slow 3? I suggest that the authors perform the analysis by breaking this band into two sub-bands of frequencies ranges (0.073 -0.1 Hz and 0.1 – 0.198 Hz). It is important to do this since if the results are significant in sub-band of 0.073 -0.1 Hz only then it may mean that the standard method of working with this frequency range should be enough and there is no need to analyze various frequency bands in case of resting-state fMRI studies.
Response 3: Thank you very much for your question. We divided slow3 into two frequency bands. And we have supplemented the recruitment integration experiment in two frequency bands(0.073-0.1Hz、0.1-0.198Hz). The results are shown in the figure below:
Figure 4. Window length = 50; The difference of recruitment and integration from 0.073-0.1Hz and 0.1-0.198Hz in the whole brain level between NC and SZ. (A) Integration. (B) Recruitment.
The results showed that the recruitment integration under the two frequency bands (0.073-0.1Hz and 0.1-0.198Hz) is insignificant. Therefore, it does not mean that a specific frequency in slow3 is substantial. Moreover, according to Zuo's article[8], the slow3 is divided uniformly, which we do not randomly select.
References
- Gifford, G., et al., Resting State fMRI Based Multilayer Network Configuration in Patients with Schizophrenia. 2020. 25: p. 102169.
- Leonardi, N. and D.J.N. Ville, On spurious and real fluctuations of dynamic functional connectivity during rest. 2015.
- Shakil, S., S.D. Keilholz, and C.H. Lee. On frequency dependencies of sliding window correlation. in IEEE International Conference on Bioinformatics & Biomedicine. 2015.
- Shakil, S., et al., Parametric Dependencies of Sliding Window Correlation. 2017. PP(99): p. 1-1.
- Mandke, K., et al., Comparing multilayer brain networks between groups: Introducing graph metrics and recommendations. 2018. 166: p. 371.
- Sba, B., et al., The structure and dynamics of multilayer networks - ScienceDirect. 2014. 544(1): p. 1-122.
- Wang, M., et al., Hierarchical Structured Sparse Learning for Schizophrenia Identification. 2019.
- Zuo, X.N., et al., The oscillating brain: complex and reliable. 2010. 49(2): p. 1432-1445.

Round 2
Reviewer 1 Report
The authors have addressed most of my concerns, but the clarity is still an issue.
The first sentence in the abstract in not understandable (i.e., a 'frequency' cannot 'focus'). Please revise at least for 'subject' and 'verb'. I think I understand what the authors mean, i.e., that the analysis of resting-state fMRI signals usually focuses in the low frequency range/band (0.01-0.1Hz) and that this does not cover all aspects of brain activity.
But I cannot give suggestions for all the sentences...
I.e., 'the brain community in schizophrenia'... it is not clear what this means.
I strongly revision by a neuroscientist fluent in English before publication.
Author Response
Response to Reviewer 1 Comments
We have carefully revised the specific details of the manuscript and be marked up using *the “Track Changes” function*. See the following WORD documents for detailed information.
